# *Toxoplasma gondii* Genetic Diversity in Mediterranean Dolphins

**DOI:** 10.3390/pathogens11080909

**Published:** 2022-08-12

**Authors:** Mercedes Fernández-Escobar, Federica Giorda, Virgina Mattioda, Tania Audino, Fabio Di Nocera, Giuseppe Lucifora, Katia Varello, Carla Grattarola, Luis Miguel Ortega-Mora, Cristina Casalone, Rafael Calero-Bernal

**Affiliations:** 1SALUVET, Department of Animal Health, Faculty of Veterinary Sciences, Complutense University of Madrid, 28040 Madrid, Spain; 2OIE Collaborating Centre for the Health of Marine Mammals, Istituto Zooprofilattico Sperimentale del Piemonte, Liguria e Valle d’Aosta, 10154 Torino, Italy; 3Istituto Zooprofilattico Sperimentale del Mezzogiorno, 80055 Portici, Italy

**Keywords:** *Toxoplasma gondii*, cetaceans, Mediterranean Sea, genotype, PCR-RFLP, PCR-sequencing

## Abstract

*Toxoplasma gondii* constitutes a major zoonotic agent but also has been frequently identified as an important cause of clinical disease (e.g., abortion, pneumonia, encephalitis) in wildlife; specifically, *T. gondii* has been associated with neurological disease in cetaceans. This study investigated the genetic diversity of *T. gondii* strains involved in infections in dolphins found stranded in the Mediterranean coastlines of Italy. Tissue samples from 16 dolphins (*Stenella coeruleoalba* and *Tursiops truncatus* species) positive for *T. gondii*-DNA presence by PCR were examined by histology and subjected to further genetic characterization of strains detected by PCR-RFLP and multilocus PCR-sequencing assays. According to fully genotyped samples, the genotypes ToxoDB#3 (67%) and #2 (22%) were detected, the latter being reported for the first time in cetaceans, along with a mixed infection (11%). Subtyping by PCR-seq procedures provided evidence of common point mutations in strains from southwestern Europe. Despite evidence of *T. gondii* as a cause of neurological disease in dolphins, sources of infections are difficult to identify since they are long-living animals and some species have vast migration areas with multiple chances of infection. Finally, the genetic diversity of *T. gondii* found in the dolphins studied in the Mediterranean coastlines of Italy reflects the main genotypes circulating inland in the European continent.

## 1. Introduction

*Toxoplasma gondii* (Apicomplexa) constitutes one of the most successful protozoan parasites worldwide and can cause harmful effects in the host, especially mammals, in which abortions and fetal malformations in pregnant people, and severe pneumonia and encephalitis in immunocompromised individuals, may be observed [1].

*Toxoplasma gondii* has a wide range of susceptible intermediate hosts, including cetaceans, in which clinical infections mostly associated to encephalitis have been frequently reported [2,3,4,5]. One of the most intriguing facts in toxoplasmosis research has to do with linking the clinical outcomes with the genotype of the strain causing the infection [6]; until date, such an association has only been observed for a few genotypes (e.g., ToxoDB#65 causing ocular disease in humans). Other aspects, such as the physiological status of the host and its genetic background, may be necessarily involved in the virulence of the *T. gondii* strains [7]. 

To date, the *T. gondii* genetic population is known to be structured in 16 well-defined haplogroups assorted into six major clades (clade A–F) worldwide. Three clonal types dominate the northern hemisphere, and a fourth clonal lineage (HG12) is largely confined to wild animals in North America. In contrast, much greater genetic diversity is observed in South America, where there seems to be no predominance of any genetic type [6]. Data on the genetic diversity of the *T. gondii* strains involved in infections of cetaceans are scarce [1]. Some genotypes predominate in certain geographical areas, but dolphins have wide-ranging areas, and their migration routes may constitute a key factor in the potential diversity of the parasite. In addition, mixed infections can be expected due to their long-life expectancy. Recently, molecular data of *T. gondii* strains infecting dolphins and other cetaceans in Europe have been summarized [8,9,10], but still a notable gap of information remains. The interest in further research on *T. gondii* in cetaceans is undeniable as an important issue in their conservation, and as potential sentinels for environmental contamination by *T. gondii* oocysts in a highly anthropized area such as the Mediterranean Sea. 

The present paper aimed at providing detailed molecular characterization (genotyping) of the *T. gondii* organisms infecting dolphins found stranded along the Italian coasts, covering new cases and others previously reported that have been revisited. 

## 2. Results

### 2.1. Parasite Detection

All sixteen cases resulted positive for *T. gondii* presence by conventional nested PCR performed during routine procedures at time of necropsy; twenty-eight target organs from the aforementioned dolphins were subjected to further analyses, based on tissue availability (Table 1). 

### 2.2. Histological and Parasite Burden Findings

Histopathological data were available for all the tissues submitted to the molecular/genotyping characterization, with the spleen of case #15 being the only exception due to logistical issues during necropsy procedures.

Eighteen tissues from 13 cetaceans showed microscopic lesions suggestive of or, at least, compatible with *T. gondii* infection (Figure 1). 

Thirteen out of the above eighteen specimens presented with mature *T. gondii*-like tissue cyst structures (12/13 brains and 1/13 skeletal muscle) (Table 1). Parasite load estimation by qPCR ranged from 0.44 to 5507.32 zoites/mg of analyzed tissues; as expected, most of the higher parasite burdens were observed in CNS specimens belonging to animals showing an apparently active *T. gondii* neurological infection (severe encephalitis) (Table 1). Some samples that tested positive by the initial nested PCR resulted negative by the subsequent quantitative PCR assay (n = 4) and it was not possible to quantify the parasite load, demonstrating the greater sensitivity of the former protocol. Association of the *T. gondii* infection with the hypothetical *causa mortis* or the cause of stranding was not among the aims of present paper. 

### 2.3. Genotyping Results

All tissues that tested positive for *T. gondii*-DNA presence by qPCR, some belonging to the same animal, were subjected to PCR-RFLP and PCR-sequencing procedures. Finally, typing was possible only for tissues from 13 different dolphins (Table 2), but a full genotyping profile was obtained only for nine animals. Genotypes ToxoDB #3 (67%, (6/9)) and ToxoDB #2 (22%, (2/9)) were observed. In addition, a mixed infection (11%, (1/9)) predictably involving type II and III strains (liver from case #4, Table 2) was observed. 

The *CS3* marker, proposed to have a highly predictive value on virulence in mice [11], presented type II alleles in all strains with the ToxoDB #3 genotype, whereas type III alleles were detected in all isolates with the ToxoDB #2 genotype (Table 2).

Regarding subtyping by multilocus PCR-sequencing procedures, we conducted PCR sequencing of three polymorphic genes, *GRA6*, *GRA7* and *SAG3*. First, all *GRA6* sequences corresponding to type III alleles (dolphin cases #13 and #16; ON814572) presented a 100% homology with MT370490 (sheep, Spain), MK055338 (cattle, Iran), MG587985 (wild boar, Italy), and many other sequences deposited in GenBank. On the other hand, all *GRA6* sequences corresponding to type II alleles (rest of dolphin cases; ON814571) showed 100% homology with MT370491 (sheep, Spain), MG587975 (pig, Italy), MG587959 (pig, Italy), and many other sequences deposited in GenBank. Concerning the *GRA7* marker, all sequences corresponding to type III alleles (dolphin cases #13 and #16; ON982169) presented a 100% homology with MT361129 (sheep, Spain), LN714496 (VEG reference strain), HQ852155 (goat, USA) and many other sequences deposited in GenBank. Moreover, all *GRA7* sequences corresponding to type II alleles (rest of dolphin cases; ON982166) showed 100% homology with MT361127 (sheep Spain), DQ459445 (PRU strain reference), JX045585 (sheep, USA) and other deposited sequences. Furthermore, the alignment of all *SAG3* sequences from samples that showed a type II allele identified a single nucleotide polymorphism (SNP), G1691T, which splits our type II and type II-like samples into two groups. The first group (dolphin cases #1, 2, 6, 9, 10 and 15; IIa *SAG3* allele, ON814568) had 100% homology with MT361125 (sheep, Spain), KU599489 (cat, Turkey), KU599478 (chicken, Portugal), ON814566 (Me49 reference strain), and others deposited in GenBank. The other group (G1691T, cases #4, 7, 11 and 14; IIb *SAG3* allele, ON814569) showed 100% identity with MT361126 (sheep, Spain), KU599488 (cat, Turkey), KU599479 (pig, Portugal), and KU599412 (sheep, France), among many other sequences deposited. The SNP leads to an amino acid change at codon 368 from Met to Ile, and this had been previously described in a large collection of samples collected from sheep abortion cases in Spain [12]. Finally, all *SAG3* sequences corresponding to type III alleles (dolphin cases #13 and #16; ON814570) presented a 100% homology with MT361130 (sheep, Spain), MK801823 (sheep, Iraq), LC414534 (rat, Iran), KU599490 (human, Turkey), and many other sequences deposited in GenBank. It should be noted the complete lack of sequences from some marine mammals infections deposited in the NCBI database.

In the case #4 (1267/15), an attempt to confirm the mixed infection by sequencing the PCR products obtained resulted in only type II allele amplification. 

### 2.4. Phylogenetic Analyses

Phylogenetic analyses were carried out aiming at positioning the strains infecting the dolphins in a One Health approach. Thus, *SAG3* marker sequences deposited in GenBank and belonging to *T. gondii* strains isolated from inland African and European hosts (representing the Mediterranean context) were included in the construction of a phylogenetic tree along with those obtained here from dolphin clinical samples and the clonal reference strains used (TgRH, TgMe49 and TgNED) (Figure 2). A separation of the selected sequences into three well-differentiated groups is evident. High bootstrap (BP) values at each node supported the clustering into A, B and C clusters (BP = 76–99%). Cluster A includes sequences with type II alleles, cluster B involves sequences with type I alleles, and finally, cluster C groups sequences with type III alleles. It should be noted that, within cluster A, although IIa and IIb along with other variants are grouped, low BP values (BP = 9–40%) indicate that the phylogenetic position of the different *Toxoplasma* strains included is not conclusive based on *SAG3* sequences used, probably due to a low diversity at the nucleotide sequence level and the short length of the *SAG3* sequences obtained.

## 3. Discussion

The study of the diseases that commonly affect cetaceans, and in particular the dolphins, is a challenging issue due to the difficult access to samples from these animals, which in most cases are derived from dead animals found stranded several hours or days after death.

Some previously investigated cases were revisited and molecularly characterized in depth here, aiming to provide reliable and comparable information of the cases and final phylogenetic analyses. In areas such as North America, clinical toxoplasmosis is a common finding in marine mammals (e.g., sea otters) [8]; in the Mediterranean basin, although there are many uncovered areas, protozoal meningoencephalitis has been linked to *T. gondii* subacute to chronic infections in dolphins [2,3,13,14,15], including through congenital transmission [16,17]. In the present study a noticeable proportion of cases (13/16–81%) presented with encephalitis and other lesions compatible with active toxoplasmosis. There is still a clear debate on the role of *T. gondii* as the cause of death in cetaceans and a predisposing factor for stranding, although in recent decades different authors have proposed its role as a primary pathogen in various stranding events [2,18,19].

In the present paper, despite the limited genetic diversity in *T. gondii* strains infecting dolphins studied, it is suggested that infections may have occurred near the coastlines of Europe where only four different genotypes (ToxoDB#1, 2, 3, 10) have been identified, along with a few recombinant and non-canonical strains [10]. This is the first report of RFLP genotype #2 in cetaceans. In addition, dolphins may be exposed to strains circulating in North Africa where a noticeable proportion of type III strains (ToxoDB#2) is present, along with a number of African genotypes, the latter being not identified in the present collection, [20]. Indeed, striped dolphins, that in our study accounted for almost the totality of the individuals (15/16–94%), consist of a species for which vast migrations within the Mediterranean basin have been hypothesized [21,22], differently for what reported for bottlenose dolphins, characterized by a residential attitude [23].

Noteworthy also is the marked susceptibility to the infection of the striped dolphin, regarded as a pelagic species; the severe disease patterns described in this study and by several other authors [2,3,14], could be related to the lack of a mutual host–parasite coevolution that exists for coastal species, such as the bottlenose dolphin, whose members are more frequently exposed to the protozoan [24].

It is well known that *T. gondii* genotypes are somehow restricted to specific regions [25]. As commented before, there are a few reports regarding the *T. gondii* genetic diversity in cetaceans and specifically in dolphins [8], and this appears to be low or limited, with most of the cases related to clonal type II lines, such as ToxoDB#1 in South Carolina, USA [26], Costa Rica [27] and Italy [28], and variants of the type II (ToxoDB#3) in Canada [29], New Zealand [30] and Italy [13]. Two reports deserve attention, the one reported by [31] described a case of fatal disseminated *T. gondii* infection by a ToxoDB#3 strain in a captive harbor porpoise (*Phocoena phocoena*) that lived in an open sea basin in Denmark, and the report by [32] that identified, by three microsatellite markers, a type II *T. gondii* strain infecting a stranded Mediterranean fin whale (*Balaenoptera physalus*) from Italy. The study of *T. gondii* strains present in other cetaceans different than dolphins will enrich the current knowledge of “marine” life cycles of *T. gondii*. 

Unfortunately, no information on the prevalence of the infection can be drawn through this study, due to the criterion of selection of the cases investigated. Future studies are warranted to obtain precious data on the epidemiology of the parasite in the Italian waters. 

The transmission pathways and the pathogenesis of the parasite in cetaceans still remain to be clarified. For those species living the offshore waters, such as striped dolphins, oocyst-contaminated wastewaters discharged from ships have been suggested as a likely source of infection [2]. However, aquatic mammals can also become infected by feeding on mussels or fish contaminated with the protozoan. A novel marine transmission pathway comprising suspended bioparticles, bio-films, small invertebrates and gastropods has been in fact recently proposed [33].

In conclusion, findings in aquatic animals of the present study are somehow a reflection of the data observed in continental (“inland”) Europe [10] and North African countries [20,34].

Unlike in the Americas [26,27,29], there are no *T. gondii* isolates from cetaceans available in Europe; such information will add important knowledge regarding intra-genotype genetic diversity and will allow us to evaluate to phenotypic traits, and allow the implementation of unified procedures [7] adding essential data for the virulence degree of such isolates.

Future investigations are warranted because of the interest of unraveling the potential (pathogenic) role of *T. gondii* in the stranding of cetaceans, and the nature of their sources and vias of infections; furthermore, parasite (*T. gondii*) isolation from cetaceans will provide valuable information both at the genomic (Whole Genome Sequencing, WGS) and phenotypic level if virulence evaluation is addressed.

## 4. Materials and Methods

### 4.1. Materials

All cases were stranded cetaceans diagnosed during routine pathological and cause-of-death assessment by the Italian stranding network of Istituti Zooprofilattici Sperimentali, veterinary public health institutions placed under the Italian Health Ministry supervision. The animals were examined and submitted to a complete post mortem examination, according to standard protocols [35].

Epidemiological (location and date of stranding) and biological data (species, sex, age class, nutritional and decomposition status) were systematically recorded (Appendix A). The animals were divided into 3 age categories (newborn–calf, juvenile–subadult and adult) based on the total body length [35,36]. The decomposition condition of the carcasses (DCC) was classified as code 1 (extremely fresh carcass, just dead), code 2 (fresh), code 3 (moderate decomposition), code 4 (advanced decomposition), or code 5 (mummified or skeletal remains) [37]. The nutritional condition state (NCC) was classified as good, moderate or poor based morphologically on anatomical parameters such as the convexity of the dorsal profile, the rib prominence and the amount of body fat.

During necropsy, tissue samples from all the major organs were collected and split into 3 aliquots for subsequent analyses: one was kept frozen at −20 °C for microbiological investigations, one at −80 °C for biomolecular analyses, and the other was preserved in neutral buffered formalin for histological investigations [9]. According to [5], when available, ten different areas from the central nervous system (CNS) were sampled and examined, including basal nuclei, thalamus, mesencephalon, pons, obex, spinal cord and frontal, parietal, occipital and cerebellar cortex. After being fixed in 10% neutral buffered formalin, tissues were embedded in paraffin, sectioned at 4 ± 2 μm, stained with haematoxylin and eosin (H&E) and examined through a light microscope.

Sixteen cases (Figure 3), resulting in being molecularly positive for *T. gondii* through a nested PCR targeting the *ITS-1* fragment [38] during routine procedures at least in one of the target organ examined (brain, lung, lymph nodes, liver, spleen, heart and muscle) (Table 1) whose tissues were easily and quickly available at the time of the present study, were retrieved and selected for the molecular analysis performed for this investigation.

Histopathological diagnostic reports of the selected tissues were retrieved and further analyzed with the focus on the microscopic changes compatible with *T. gondii* infection [4,5,30,39].

A few cases had been previously published and revisited here (Appendix A).

### 4.2. DNA Extraction from Tissues

DNA was purified from 25 mg of frozen tissue specimens, based on tissue availability and previous positive detection to the protozoon; samples were first homogenized using the TissueLyser II (QIAGEN, Hilden, Germany) by high-speed shaking in Eppendorf tubes with stainless steel beads (5 mm diameter, QIAGEN). Homogenates were centrifuged at 14,000 rpm for 3 min to remove the suspended solids, without removing beads. Supernatants were submitted to DNA extraction using the ReliaPrep TM gDNA Tissue Miniprep System kit (Promega Corporation, Madison, WI, USA) as described by the manufacturer (Standard Protocol for Animal Tissue). The genomic DNA was placed at −20 °C for long-term storage after quantification with VivaSpec Spectrophotometer (Sartorius Stedim Biotech, Aubagne, France).

### 4.3. Parasite Quantification in Tissues

Trying to link the severity of histological lesions with the parasite burden, *T. gondii* DNA quantification was performed using a duplex qPCR assay adapted from [40]. It included the amplification of the species-specific *529RE* locus and an internal amplification control (IAC) to aid the identification of false negative results [41]. The qPCR reactions were performed in a final volume of 25 μL using the SensiFAST Probe Lo-ROX Kit (Bioline, Memphis, TN, USA; BIO-84020); each primer was at a final concentration of 0.25 μM, HEX (*529RE* locus) and Cy5 (IAC) probes were at a final concentration of 0.15 μM, as well as 5 μL of DNA. Amplification and fluorescence detection were performed on an Applied Biosystems 7500 FAST Real-Time PCR System (Applied Biosystems, Foster City, CA, USA) using 96-well PCR plates under the following conditions: initial denaturation at 95 °C for 5 min, followed by 45 cycles of 95 °C for 15 s and 60 °C for 40 s. 

Quantification (number of *T. gondii* parasites) was calculated by interpolating the average Ct values on a standard curve equivalent to 1 × 10^5^ − 1 × 10^−1^ tachyzoites generated by tenfold serial dilutions of parasite DNA. Standard curves for *T. gondii* showed an average slope always close to −3.3 and an R^2^ > 0.98. Parasite load in tissues was expressed as the zoites/mg of tissue.

### 4.4. Molecular Characterization-Genotyping of T. gondii Strains (Organisms)

All qPCR-positive samples were subjected to further genotyping analysis. DNA extracts were subjected to the widely used Mn-PCR restriction fragment length polymorphism (RFLP) method, with the markers *SAG1*, *SAG2* (*5′–3′ SAG2*, and *alt. SAG2*), *SAG3*, *BTUB*, *GRA6*, *c22-8*, *c29-2*, *L358*, *PK1*, and *Apico* [42].

Aiming for a deeper genetic characterization of the *T. gondii* population detected in dolphins, other cases of infection reported earlier (Appendix A), have been revisited. ToxoDB RFLP genotype was identified according to http://toxodb.org/toxo/ accessed on 25 April 2022. 

In addition, alleles for the *CS3* marker [11], which are suggested to have highly predictive value for the virulence degree in mice, and those from the well-known virulence gene *GRA7* [43] were also studied. In these cases, the methodology was based on nested PCR-DNA sequencing and in silico digestion of each locus sequences obtained. All details are described in [44]. The sequencing procedures were carried out at the Center for Genomic Technologies of the Complutense University of Madrid (Spain) using the BigDye^®^ Terminator kit v 3.1 (Applied Biosystems, Foster City, CA, USA) and analyzed on an ABI 3130 Genetic Analyzer (Applied Biosystems). The resulting sequences were imported, read, edited manually if necessary, and analyzed using BioEdit software (version 7.0.5.3; [https://www.bioedit.software.informer.com], accessed on 10 August 2022) [45]. Necessary alignments were performed using Clustal Omega software [https://www.ebi.ac.uk/Tools/msa/clustalo/, accessed on 1 June 2022]. Finally, in silico digestion was conducted by the NEBCutter 2.0 program [46].

### 4.5. Phylogenetic Analyses

A phylogenetic tree was constructed based on *SAG3* sequences obtained from cetacean samples included in the present study in addition to those from a set of sequences deposited in GenBank selected based on their geographical origin (Europe and Africa) in order to represent the Western Mediterranean context. Sequences from the clonal reference strains TgRH (type I), TgMe49 (type II) and TgNED (type III) were also included. The evolutionary history was inferred using the Neighbor-Joining method [47]. A bootstrap test (10,000 replicates) to evaluate the percentage of replicate trees in which the associated taxa clustered together was conducted [48]. The evolutionary distances were computed using the Maximum Composite Likelihood method [49]. Evolutionary analyses were conducted in MEGA11 [50].

## 5. Conclusions

*Toxoplasma gondii* is frequently found infecting stranded wild dolphins, and such infections are frequently associated to neuropathy; in addition, despite the limited genetic diversity found in the dolphins studied in the Mediterranean coastlines of Italy, findings reflect the main genotypes circulating inland in the European continent. Indeed, the first detection of ToxoDB genotype #2 in a cetacean in this study can be highlighted.

There are still many gaps that should be covered, such as: (i) how frequent are dolphin infections in nature, and (ii) what are the main sources and vias of infection for cetaceans. Therefore, it is expected that *T. gondii* isolation from free-ranging dolphins (and other cetaceans), and the use of tools with higher resolution power (WGS and other next generation sequencing-based methods) will provide evidence of the sources of infections and the similarity of *T. gondii* strains in dolphins with those strains found inland.

## Figures and Tables

**Figure 1 pathogens-11-00909-f001:**
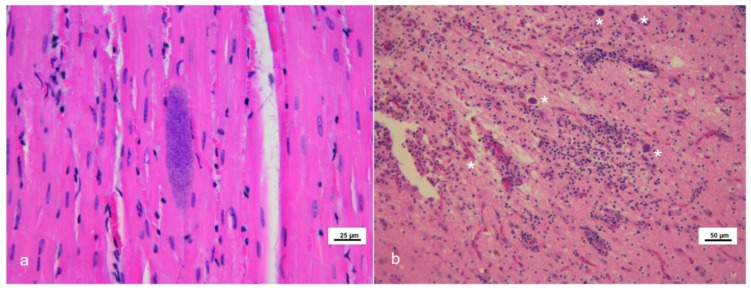
Microscopic lesions in stranded cetaceans with *Toxoplasma gondii* infection. Hematoxylin and eosin (HE) staining. (**a**) Skeletal muscle (case #6). *Toxoplasma*-like tissue cyst. (**b**) Brain frontal cortex (case #7). Severe non-suppurative necrotizing encephalitis in the presence of several *Toxoplasma*-like tissue cysts (asterisks).

**Figure 2 pathogens-11-00909-f002:**
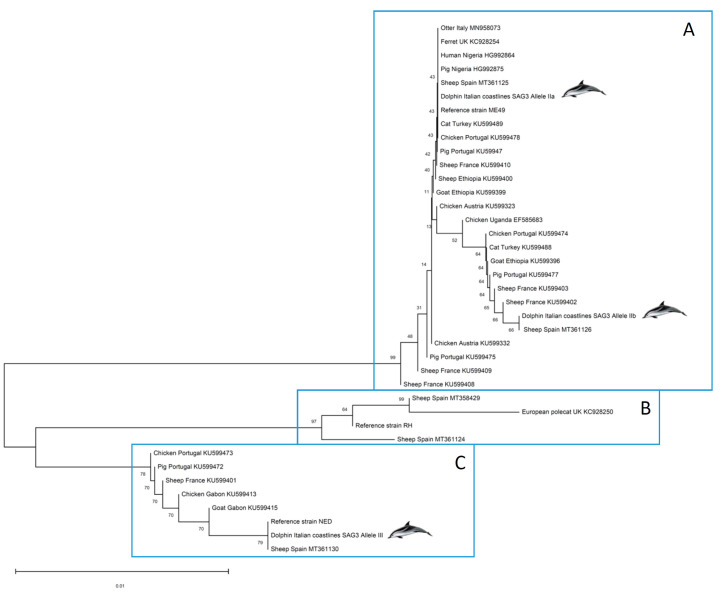
Phylogenetic positioning of the *Toxoplasma gondii* organism found in stranded dolphins tissues based on the *SAG3* gene. This analysis involved 39 nucleotide sequences from *T. gondii* strains/isolates infecting human, domestic and wild hosts located in Europe and Africa.

**Figure 3 pathogens-11-00909-f003:**
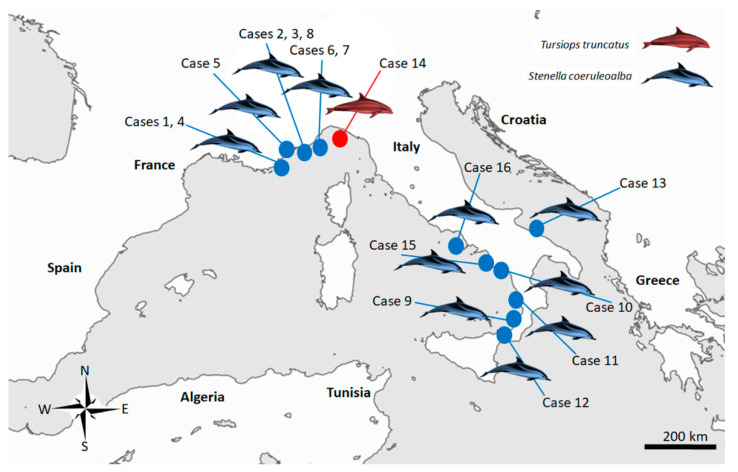
Map of the study area in the Italian Mediterranean coastline, displaying the stranding locations of the 16 cetaceans infected with *Toxoplasma gondii* selected for present study.

**Table 1 pathogens-11-00909-t001:** Histological and parasite load findings in tissue samples from dolphins stranded along the Italian coastlines that tested positive to *Toxoplasma gondii* DNA presence.

Case# (ID)	Tissue Samples Tested for *T. gondii* during Necropsy Procedures *	Histological Examination			Parasite Load by qPCR (Zoites/mg) **
		Tissue Submitted to Molecular Characterization	H&E Lesions Compatible with *T. gondii* Infection	*T. gondii*-like Tissue Cysts Observed? (Yes/No)	
#1 (547/15)	**CNS**, **lung**, heart, **liver** ^a^, **prescapular LN**, skeletal muscle ^b^	CNS	Moderate NS encephalitis	Y	28.60
		Prescapular LN	None	N	13.21
		Lung	Hemorrhages	N	103.01
#2 (3908/15)	CNS, **lung**, **spleen**, liver, prescapular LN, skeletal muscle	Lung	None	N	8.50
		Spleen	None	N	0.44
#3 (40548/15)	CNS, lung, **spleen**, liver, **heart** ^a^, **prescapular LN** ^a^, skeletal muscle	Spleen	None	N	Not detected
#4 (1267/15)	**CNS**, heart, spleen, **liver**, **prescapular LN**, skeletal muscle	CNS	Severe NS meningoencephalitis	Y	3.27
		Liver	Foci of necrosis	N	16.56
		Prescapular LN	None	N	0.83
#5 (14879/16)	**CNS**, spleen, liver, prescapular LN, heart. skeletal muscle	CNS	Moderate NS meningoencephalitis	Y	3.34
#6 (16769/17)	**CNS**, heart, **spleen**, **liver**, **bronchial** and **pulmonary LN, skeletal muscle**	CNS	Moderate NS meningoencephalitis	Y	167.40
		Liver	Foci of necrosis associated with mixed inflammatory infiltrate	N	5.76
		Pulmonary LN	Necrotizing lymphadenitis	N	4.18
		Skeletal muscle	NS myositis	Y	1.84
#7 (78983/17)	**CNS**, **heart** ^a^, spleen, liver, prescapular and bronchial LN, skeletal muscle	CNS	Severe NS necrotizing meningoencephalitis	Y	17.95
#8 (50099/18)	Liver, **prescapular** and **bronchial** ^a^ **LN**, spleen, muscle	Prescapular LN	None	N	Not detected
#9 (62728/18)	**CNS**, lung, heart, spleen, liver, prescapular LN, skeletal muscle	CNS	Mild NS meningoencephalitis	N	6.56
#10 (92929/18)	**CNS**	CNS	Severe NS meningoencephalitis	Y	83.13
#11 (95661/19)	**CNS**, lung, heart, spleen, liver, thymus, prescapular LN, skeletal muscle	CNS	Severe NS necrotizing meningoencephalitis	Y	16.21
#12 (24676/20)	**CNS, spleen** ^a^, **prescapular LN** ^a^	CNS	Moderate NS meningoencephalitis	Y	1.99
#13 (38325/20)	**CNS**, spleen, kidney, prescapular LN	CNS	Severe NS encephalitis	Y	5507.32
#14 (51352/20)	**CNS**, **heart** ^a^, **spleen**, mesenteric and **pulmonary LN**, **skeletal muscle**	CNS	Mild NS meningoencephalitis	Y	23.58
		Skeletal muscle	None	N	6.96
		Pulmonary LN	None	N	Not detected
		Spleen	None	N	0.03
#15 (2564/21)	**CNS**, lung, **spleen**, liver, prescapular LN, skeletal muscle	CNS	Severe pyogranulomatous encephalitis associated with mild meningitis	Y	316.06
		Spleen	NP		Not detected
#16 (24621/21)	**CNS**, lung, heart, spleen, liver, prescapular LN, skeletal muscle	CNS	Severe NS meningoencephalitis	Y	1537.23

NS = non-suppurative; LN = lymph node; NP = not performed; * In bold are those samples that tested PCR positive for *Toxoplasma gondii* DNA [9]. ** Specimens with parasite burden of less than 25 zoites/mg are considered untypable due to the sensitivity limits of the multiplex-PCR employed for RFLP based-genotyping, and therefore were not subjected to further molecular analyses. ^a^ Not available at the step of parasite load quantification. ^b^ Skeletal muscle examined in all cases corresponded to *longissimus dorsi*.

**Table 2 pathogens-11-00909-t002:** Results of PCR-RFLP genotyping analyses carried out on *Toxoplasma gondii* strains identified in the dolphins stranded along the Italian coastlines.

Case# (ID)	Tissue *	*SAG1*	*3′-SAG2*	*5′-SAG2*	*Alt. SAG2*	*SAG3*	*BTUB*	*GRA6*	*c22-8*	*C29-2*	*L358*	*PK1*	*Apico*	*GRA7*	*CS3*	ToxoDB #
Reference strain RH	-	I	I/III	I/II	I	I	I	I	I	I	I	I	I	I	I	#10
Reference strain Me-49	-	II/III	II	I/II	II	IIa	II	II	II	II	II	II	II	II	II	#1
Reference strain NED	-	II/III	I/III	III	III	III	III	III	III	III	III	III	III	III	III	#2
#1 (547/15)	CNS	II/III	II	I/II	II	IIa	II	II	II	II	II	II	I	II	II	#3
#2 (3908/15)	Lung	-	II	-	-	IIa	-	-	-	-	-	-	I	-	-	Likely #3
#4 (1267/17)	Liver	II/III	II	I/II	II + III	IIb	II	II	II + III	II	II	II	I	II	II	Likely mixed
#6 (16769/17)	CNS	II/III	II	I/II	II	IIa	II	II	II	II	II	II	I	II	II	#3
#7 (78983/17)	CNS	II/III	II	I/II	II	IIb	II	II	II	II	II	II	I	II	II	#3
#9 (62728/18)	CNS	-	-	-	-	IIa	-	-	II	-	-	-	I	-	-	Likely #3
#10 (92929/18)	CNS	II/III	II	I/II	II	IIa	II	II	II	II	II	II	I	II	II	#3
#11 (95661/19)	CNS	II/III	II	I/II	II	IIb	II	II	II	II	-	II	I	II	II	Likely #3
#13 (38325/20)	CNS	II/III	I/III	III	III	III	III	III	III	III	III	III	III	III	III	#2
#14 (51352/20)	CNS	II/III	II	I/II	II	IIb	II	II	II	II	II	II	I	II	II	#3
#15 (2564/21)	CNS	II/III	II	I/II	II	IIa	II	II	II	II	II	II	I	II	II	#3
#16 (24621/21)	CNS	II/III	I/III	III	III	III	III	III	III	III	III	III	III	III	III	#2

* In the table, only result of the tissue DNA from which more molecular markers were amplified within the same animal is represented. No inconsistencies in the profile obtained between tissues were observed.

## Data Availability

DNA sequences of the markers *SAG3* (ON814566-70), *GRA6* (ON814571, ON814572), and *GRA7* (ON982166-70) have been deposited in GenBank.

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
