# Peer review of "Toxoplasma gondii Genetic Diversity in Mediterranean Dolphins"

_pathogens, 2022, doi:10.3390/pathogens11080909_

Round 1

Reviewer 1 Report

This study will provide important information of genetic diversity in difficult to find samples. As the authors mentioned about microscopic lesions suggestive compatible with Toxoplasma gondii infection and tissue cyst-like structures, it would be better if the authors can add some pictures of them. Apart from T. gondii examination, the details of post mortem examination could be briefly described.

Author Response

This study will provide important information of genetic diversity in difficult to find samples. As the authors mentioned about microscopic lesions suggestive compatible with Toxoplasma gondii infection and tissue cyst-like structures, it would be better if the authors can add some pictures of them. Apart from T. gondii examination, the details of post mortem examination could be briefly described.

Authors’ response: Authors are greateful for reviewer’s kind words. A new figure 1 has been added describing the most common histological findings (tissue-cysts structures and severe non-suppurative encephalitis) observed in the animals. In addition the details of the post mortem examination are described in an ad hoc paragraph, from lines 264 to 272. A previous paper of the authors has been added as reference at line 267 to clarify better which pathogens and analyses are routinely searched and carried out, respectively.

Reviewer 2 Report

This paper uses histology and PCR based (qPCR, RFLP and MLST) techniques  to determine the genotypes of the protozoan parasite Toxoplasma gondii found in Mediterranean dolphins. The genetypes found in the dolphins reflect those found on mainland Euope. Given that toxoplasmasosis causes defects in developing foeteuses and neurological symptoms this study provides an important basis for further study to determine the relative pathogenesis of T. gondii genotypes on doplhins and subsequent risk to health of dolphin populations and association with strandings.

Overall it is well written, although there are some minor English language nuances that are highlighted below.

Three major points that must be addressed are:

1)  the DNA sequences generated and used in this study must be downloaded onto GeneBank and the accession numbers given in the paper before publication.

2) The introduction requires greater detail about T. gondii genotypes to put work performed in this paper into context.

3) You need to make it clear why you selected these 16 carcasses for testing and how many carcasses were tested in the first place to identify the 16 positives.

Minor corrections:

Abstract

Line 18: 'associated to neurological..' change to 'associated with neurological...'

lines 18-19: 'Present article aims at studying the...' should be changed to 'This study investigates...'

Lines 23-24: it is unclear whether the percentages of genotypes stated are asosciated with individual cases or with some other factor. For example the mixed infection appears to be just one dolphin out of 16, which you would assume is 6.25% not 11%, so please check and make it clear what the %'s refer to. The 11% stated re mixed infection may refer to the 1 dolphin sample of the 9 that underwent full genotyping profiles...but that should be made clear otherwise readers will assume that 11% of all samples tested were mixed. Same point for the other two percentages stated.

Line 28: 'Finally, genetic diversity found in the doplhins...' change to 'Finally, genetic diversity of T. gondii found in the doplphins...

Line 29: 'reflect the main genotypes...' change to 'reflects the main genotypes...'

Introduction

Would benefit from further information about Toxoplasma genotypes to put the subsequent work and results into context. Present level of information is not sufficient.

Lines 56-61: you are basically repeating the same information here in your aims and your results. Merge these together and state as aims in your introduction.

Results

Line 65: 'on tissues...' change to 'on tissue...'

Lines 95-96: 'Finally, strains found infecting tissues from 13 different dolphins were typed...' this sounds as if you have at this stage introduced 13 brand new dolphin carcasses into the study in addition to the 16 you started with. Make it clear that you mean that T. gondii strains from 13 of the 16 dolphins sampled were then genotyped.

Figure 1

It would be beneficial to show the ToxoDB genetypes associated with A, B and C on the image, and / or mention which genotypes are associated with which cluster in the figure legend.

Discussion

Line 188: this makes it sound as if you are studying the genetic diversity of the dophins, consider changing the text to make it clear you are referring to T. gondii genetype diversity.

Line 199: 'Noteworthy also the marked...' change to 'Noteworthy also is the marked....'

Line 204: consider changing 'circumscribed' to 'restricted'.

Lines 216-217: 'are still far to be clarified.' change to 'still remain to be clarified.'.

Materials and Methods

Major point: It is unclear whether 16 dolphin carcasses sampled were the only carcasses sampled or whether you sampled more but only 16 of these were positive for T. gondii. Make it clear how many doplhin carcasses were investigated in total, and what percentage of these were found to be T. gondii positive, as this provides useful information as to the prevalnece of infection within dolphin populations.  Otherwise it implies that 100% of dolphins carcasses in the Mediterranean are positive for T. gondii. If you do not have this data, then you need to state what your criteria for selecting these carcasses was.

Lines 289-293: clarify whether the qPCR reaction volume was 20 or 25 microlitres. Both are stated.

Lines 307-309: it is unclear what other cases are being referred to in table 2 as it appears to be 3 reference strains and 12 samples which are derived from the 16 samples originally tested.

References

Ensure ALL latin binomials are in itallics i.e. line 383 Toxoplasma gondii

Author Response

Comments and Suggestions for Authors

This paper uses histology and PCR based (qPCR, RFLP and MLST) techniques to determine the genotypes of the protozoan parasite Toxoplasma gondii found in Mediterranean dolphins. The genetypes found in the dolphins reflect those found on mainland Euope. Given that toxoplasmasosis causes defects in developing foeteuses and neurological symptoms this study provides an important basis for further study to determine the relative pathogenesis of T. gondii genotypes on doplhins and subsequent risk to health of dolphin populations and association with strandings.

Overall it is well written, although there are some minor English language nuances that are highlighted below.

Three major points that must be addressed are:

1) the DNA sequences generated and used in this study must be downloaded onto GeneBank and the accession numbers given in the paper before publication.

2) The introduction requires greater detail about T. gondii genotypes to put work performed in this paper into context.

3) You need to make it clear why you selected these 16 carcasses for testing and how many carcasses were tested in the first place to identify the 16 positives.

Authors’ response: Authors really appreciate reviewer’s kind words and accurate comments and English language has been revised. Firstly, all GenBank numbers have been added to the current version of the manuscript version. Secondly, some lines have been added to the introduction in order to put the reader into context about T. gondii genetic diversity worldwide. Finally, regarding the selection of the cases, sixteen is the number of animals resulting molecularly positive to T. gondii (routine analysis performed to each animal stranded and analyzed by the Italian national reference center) and whose tissues were easily and quickly available for this study. For an exhaustive explanation please see what reported below, in the section referring to the materials and methods. The text has been modified accordingly.

Minor corrections:

Abstract

Line 18: 'associated to neurological..' change to 'associated with neurological...'

Authors’ response: Changes were made as suggested.

lines 18-19: 'Present article aims at studying the...' should be changed to 'This study investigates...'

Authors’ response: Changes were made as suggested.

Lines 23-24: it is unclear whether the percentages of genotypes stated are asosciated with individual cases or with some other factor. For example the mixed infection appears to be just one dolphin out of 16, which you would assume is 6.25% not 11%, so please check and make it clear what the %'s refer to. The 11% stated re mixed infection may refer to the 1 dolphin sample of the 9 that underwent full genotyping profiles...but that should be made clear otherwise readers will assume that 11% of all samples tested were mixed. Same point for the other two percentages stated.

Authors’ response: Authors apologize for the misunderstanding. Percentages are associated with the number of fully genotyped animals (n=9), because only in the case of fully typed cases a genotype could be reliably designed. A sentence has been added to clarify that.

Line 28: 'Finally, genetic diversity found in the doplhins...' change to 'Finally, genetic diversity of T. gondii found in the doplphins...

Authors’ response: Changes were made as suggested.

Line 29: 'reflect the main genotypes...' change to 'reflects the main genotypes...'

Authors’ response: Changes were made as suggested.

Introduction

Would benefit from further information about Toxoplasma genotypes to put the subsequent work and results into context. Present level of information is not sufficient.

Authors’ response: As indicated above, some lies have been added to the introduction in order to put the reader into context about T. gondii genetic diversity worldwide.

Lines 56-61: you are basically repeating the same information here in your aims and your results. Merge these together and state as aims in your introduction.

Authors’ response: Changes were made as suggested.

Results

Line 65: 'on tissues...' change to 'on tissue...'

Authors’ response: Changes were made as suggested.

Lines 95-96: 'Finally, strains found infecting tissues from 13 different dolphins were typed...' this sounds as if you have at this stage introduced 13 brand new dolphin carcasses into the study in addition to the 16 you started with. Make it clear that you mean that T. gondii strains from 13 of the 16 dolphins sampled were then genotyped.

Authors’ response: Changes were made as suggested.

Figure 1

It would be beneficial to show the ToxoDB genetypes associated with A, B and C on the image, and / or mention which genotypes are associated with which cluster in the figure legend.

Authors’ response: Please note that now this figure will appear as Figure 2. Thank you for the suggestion, nevertheless adding the ToxoDB number is not fully correct because the genotype # cannot be inferred from the analysis of a single genetic marker. The association to one of the groups was a way to represent and “allocate” other genetic findings in the Mediterranean context. This is clear that for most of the SAG3 sequences deposited in GenBank such are not associated to a full RFLP profile in their original papers, what makes this task as impossible.

Discussion

Line 188: this makes it sound as if you are studying the genetic diversity of the dophins, consider changing the text to make it clear you are referring to T. gondii genetype diversity.

Authors’ response: Thank you for the remark. The text has been revised accordingly.

Line 199: 'Noteworthy also the marked...' change to 'Noteworthy also is the marked....'

Authors’ response: Changes were made as suggested.

Line 204: consider changing 'circumscribed' to 'restricted'.

Authors’ response: Changes were made as suggested.

Lines 216-217: 'are still far to be clarified.' change to 'still remain to be clarified.'

Authors’ response: Changes were made as suggested.

Materials and Methods

Major point: It is unclear whether 16 dolphin carcasses sampled were the only carcasses sampled or whether you sampled more but only 16 of these were positive for T. gondii. Make it clear how many doplhin carcasses were investigated in total, and what percentage of these were found to be T. gondii positive, as this provides useful information as to the prevalnece of infection within dolphin populations. Otherwise it implies that 100% of dolphins carcasses in the Mediterranean are positive for T. gondii. If you do not have this data, then you need to state what your criteria for selecting these carcasses was.

Authors’ response: Thank you for such important remark. Unfortunately, through this study, we cannot derive information regarding the prevalence of the infection in cetaceans stranded along the Italian coasts (although this would be very interesting), because the sixteen selected cases correspond to animals that were molecularly positive for T. gondii, whose matrices were easily and quickly available at the time of this study and do not correspond to the totality of the subjects that tested positive for the protozoan. Thanks for pointing this out. To clarify this concept, we added the sentence “whose tissues were easily and quickly available at the time of the present study” at line 276 in materials and methods section. In addition, a new paragraph has been added in discussion; please see lines 224-227.

Lines 289-293: clarify whether the qPCR reaction volume was 20 or 25 microlitres. Both are stated.

Authors’ response: Authors apologize for the mistake. Changes were made.

Lines 307-309: it is unclear what other cases are being referred to in table 2 as it appears to be 3 reference strains and 12 samples which are derived from the 16 samples originally tested.

Authors’ response: Authors apologize for the mistake. Authors meant to refer to the Supplementary Table S1 instead of Table 2. Supplementary file summarizes the new and revisited cases.

References

Ensure ALL latin binomials are in itallics i.e. line 383 Toxoplasma gondii

Authors’ response: Authors apologize for the multiple mistakes. Changes were made.

Reviewer 3 Report

This is a thorough study of Toxoplasma in dolphins in Italy. Proper methods were used,Line and results are broad scientific interest. I have only a few minor suggestions.

Line 52-replace needs to be filled with remains

Line 73-delete immature, like not in italics

Line 175-replace come from with derived

Line 180-add some before marine

line 219-get the infection replace with become

line 223-replace sum with conclusio

Author Response

This is a thorough study of Toxoplasma in dolphins in Italy. Proper methods were used, Line and results are broad scientific interest. I have only a few minor suggestions.

Authors’ response: Authors really appreciate reviewer’s kind words and accurate comments.

Line 52-replace needs to be filled with remains

Authors’ response: Changes were made as suggested.

Line 73-delete immature, like not in italics

Authors’ response: Changes were made as suggested.

Line 175-replace come from with derived

Authors’ response: Changes were made as suggested.

Line 180-add some before marine

Authors’ response: Changes were made as suggested.

line 219-get the infection replace with become

Authors’ response: Changes were made as suggested.

line 223-replace sum with conclusio

Authors’ response: Changes were made as suggested.